# Prediction of the Number of Cumulative Pulses Based on the Photon Statistical Entropy Evaluation in Photon-Counting LiDAR

**DOI:** 10.3390/e25030522

**Published:** 2023-03-17

**Authors:** Mingwei Huang, Zijing Zhang, Longzhu Cen, Jiahuan Li, Jiaheng Xie, Yuan Zhao

**Affiliations:** School of Physics, Harbin Institute of Technology, Harbin 150001, China; 18B911040@stu.hit.edu.cn (M.H.); 1121120119@hit.edu.cn (L.C.); lijiahuan@hit.edu.cn (J.L.); xiejiaheng@hit.edu.cn (J.X.)

**Keywords:** photon-counting LiDAR, photon statistical entropy, signal evaluation, prediction model

## Abstract

Photon-counting LiDAR encounters interference from background noise in remote target detection, and the statistical detection of the accumulation of multiple pulses is necessary to eliminate the uncertainty of responses from the Geiger-mode avalanche photodiode (Gm-APD). The cumulative number of statistical detections is difficult to select due to the lack of effective evaluation of the influence of the background noise. In this work, a statistical detection signal evaluation method based on photon statistical entropy (PSE) is proposed by developing the detection process of the Gm-APD as an information transmission model. A prediction model for estimating the number of cumulative pulses required for high-accuracy ranging with the background noise is then established. The simulation analysis shows that the proposed PSE is more sensitive to the noise compared with the signal-to-noise ratio evaluation, and a minimum PSE exists to ensure all the range detections with background noise are close to the true range with a low and stable range error. The experiments demonstrate that the prediction model provides a reliable estimation of the number of required cumulative pulses in various noise conditions. With the estimated number of cumulative pulses, when the signal photons are less than 0.1 per pulse, the range accuracy of 4.1 cm and 5.3 cm are obtained under the background noise of 7.6 MHz and 5.1 MHz, respectively.

## 1. Introduction

Photon-counting LiDAR, equipped with a Geiger-mode avalanche photodiode (Gm-APD), has a high capability for photon-efficient sensitivity and picosecond-level time resolution and the potential for range detection and 3D imaging for remote targets [1,2,3,4,5,6,7]. Affected by the probabilistic response of the Gm-APD, the detection process of photon-counting LiDAR is statistical, which requires the accumulation of multiple pulses to locate the echo signal [8,9,10]. Nevertheless, in scenarios with strong background noise, the selection of the number of cumulative pulses is a common problem. The main reason for this is the lack of an effective metric to evaluate the quality of the statistical detection.

In detection theory, the signal-to-noise ratio (SNR) is a universal metric for evaluating signal quality. Since Pellegrini et al. introduced SNR into photon-counting LiDAR [11], it has been widely used in the signal quality assessment of photon-counting statistical detection. Researchers have proposed various schemes to define the SNR according to the different application requirements and the system’s characteristics, e.g., a definition from the perspective of a photon-counting histogram [12], a definition based on the continuous time at which the detector responds to the first echo photon [13], and a definition given by the number of signal photoelectrons and noise photoelectrons [14]. These schemes follow the same formula: the SNR is equal to the ratio of the mean of the signal to the Poisson fluctuation. Mccarthy et al. used the defined SNR to analyze a scanning LiDAR prototype and build a prediction model for photon-counting LiDAR under different working environments. The prediction model determined the minimum SNR required to reliably lock on to the actual target range, estimated the maximum resolvable range under different dwell times, and analyzed the detection performance for underwater targets [15,16,17]. At present, the SNR metrics in photon-counting LiDAR are limited to the low-flux region, and the abovementioned research used SNR to evaluate the signal quality in the scenes where the return counting rate was no more than 10 kHz (or counts/s) [13,14]. The applicability of the SNR metrics in a high background-noise environment (up to MHz level) remains to be tested. Additionally, another commonly used evaluation metric in photon-counting statistical detection is the so-called signal-to-background ratio (SBR) [18,19,20], whose definition is the ratio of the mean of the signal to the mean of the background noise. Unfortunately, the SBR cannot indicate the influence of the number of cumulative pulses on the signal quality of the statistical detection.

In fact, detection probability is the most objective metric to evaluate the signal quality of photon-counting statistical detection, which is determined by the nonlinear response characteristics of the Gm-APD; that is, the signal detection probability is affected by all the noise arriving before the signal [21,22]. Based on the analysis of the detection probability of the Gm-APD, Gatt et al. proposed an SNR scheme with a detection probability and analyzed the influence of the detectors’ deadtime and noise rate on the SNR in statistical detection [23]. However, it is more important for the detection probability to be analyzed in each time bin, which is difficult to achieve in photon-counting LiDAR for remote targets because the target’s range is always unknown. Therefore, this version of SNR is not convenient in the practical application.

To summarize, an effective signal quality evaluation method needs to be developed for photon-counting LiDAR working in background noise. Entropy involves higher-order statistics compared with the conventional counting statistics and probability statistics. It is an important quantitative tool to evaluate the uncertainty of information generation, transmission, and reception. In recent years, many studies using entropy as an evaluation metric to improve performance have emerged in various fields, such as image quality assessment [24,25], beam complexity analysis [26], laser chaotic signal evaluation [27,28], fuzzy clustering [29,30], quantum communication [31,32], etc. In photon-counting LiDAR, the process of range measurement can be regarded as an information transmission model, and the uncertainty of the distribution of the signal and noise is completely different. The noise is uniformly distributed with high uncertainty, while the signal is distributed in pulses with lower uncertainty. Therefore, it is possible to utilize entropy as a metric to analyze the uncertainty of the distribution of the echo photons for signal evaluation in photon-counting LiDAR.

In this work, with the analysis of the detection probability of the Gm-APD, a signal quality evaluation method for photon-counting LiDAR based on photon statistical entropy (PSE) is proposed; then, a prediction model for determining the number of cumulative pulses in high background-noise scenarios is established. Specifically, the detection process of the Gm-APD is developed as an information transmission model, and the PSE is proposed as the signal evaluation metric. Additionally, the number of required cumulative laser pulses for targets’ ranging with background noise is estimated using the minimum PSE reference. The simulation and experiment demonstrate that compared with the SNR evaluation in statistical detection, the proposed PSE metric is more sensitive to indicating the influence of the noise, and the PSE-based prediction model provides a good estimation of the number of cumulative pulses required for high-accuracy ranging for photon-counting LiDAR working in MHz-level background noise.

## 2. PSE Evaluation in Photon-Counting LiDAR

### 2.1. Forward Probability Model in Photon-Counting LiDAR

In the photon-counting LiDAR system, often, a pulsed laser is triggered to illuminate the target, a Gm-APD is synchronized to respond to the echo photons reflected from the target, and a time-to-digital module is used to record and output the time difference between the emission and the arrival of the laser pulse. For typical Gaussian laser pulses, the echo optical flux at time *t* is expressed as
(1)$$s\left(t|\tau \right)=N_{s}\frac{1}{\sqrt{2\pi }\sigma }\exp\left(-\frac{{\left(t-\tau \right)}^{2}}{2\sigma^{2}}\right)+f_{n},$$
where
$$N_{s}=\frac{E_{t}}{h\nu }\frac{A_{r}}{2\pi R^{2}}\rho \eta_{r}\tau_{a}^{2}$$
denotes the number of echo signal photons [33,34], which is determined by the single-pulse energy $E_{t}$; the single photon energy hν, defined by the Planck constant *h* and the laser frequency ν; the receiving system’s area $A_{r}$ and transmittance $\eta_{r}$; the target’s reflectivity ρ and range *R*; and the transmittance of atmosphere $\tau_{a}$. $f_{n}$ denotes the total counting rate of the noise that consists of the background noise and the dark count. σ is the mean square root of the Gaussian pulse. τ=2R/c is the time of flight of the laser pulses, in which *c* is the speed of light. In a practical system, the variate *t* is often discretized into time bins, with a width of Δt for each bin. In the *j*th bin, the forward probability, i.e., the probability that the Gm-APD responds to the echo photons, is
(2)$$\begin{array}{cc} P\left(j|i\right)= & \exp\left(-\int_{0}^{(j-1)\Delta t}\eta s\left(t|\tau \right)dt\right)\\ \; & \times \left(1-\exp\left(-\int_{\left(j-1\right)\Delta t}^{j\Delta t}\eta s\left(t|\tau \right)dt\right)\right), \end{array}$$
where η is the quantum efficiency of the Gm-APD; i=τ/Δt is the *i*th bin after the same discretization to τ; and i=1,⋯,N,j=1,⋯,N, in which $N=T_{g}/\Delta t$, the number of time bins, is determined by the width of range gate $T_{g}$. Equation (Equation 2) is derived from Poisson statistics [22], and it explains the response probability in the *j*th bin, when the target is located in the *i*th bin. The response of the Gm-APD probabilistically occurs in every possible bin due to the influence of noise. Assuming that there exists only one target in the scene, the forward probability matrix that indicates the capacity of photon-counting statistical detection is obtained by the traversal of all *i* and *j*
(3)$$P_{t|\tau }=\left[P\left(j|i\right)\right],i=1,\;\cdots ,\;N\;\mathrm{and}\;j=1,\;\cdots ,\;N,$$
where the square bracket means $P_{t|\tau }$ is a matrix. Equation (Equation 3) describes the response probability of the Gm-APD in each bin, with the consideration of all possible cases of the target location in the range gate. Based on Equations (Equation 2) and (Equation 3), the detection process of the photon–electron conversion of the Gm-APD is built as an information transmission model.

### 2.2. PSE for Single-Pulse Detection

On the foundation of Equation (Equation 3), the PSE for single-pulse detection is defined in the form of mutual information [35]
(4)$$I_{S}=E_{P_{t,\tau }}\left(\log P_{t,\tau }-\log P_{\tau }{\left(P_{t}\right)}^{T}\right),$$
where $P_{t,\tau }=\left[P\left(j,i\right)\right]$ denotes the joint probability of the target location and the distribution of the Gm-APD’s responses; $E_{P_{t,\tau }}\left(\cdot \right)$ is the expectation operation of the joint probability $P_{t,\tau }$; $P_{\tau }={\left[P\left(i\right)\right]}^{T}$ denotes the probability of the target location or the prior probability; and $P_{t}={\left[P\left(j\right)\right]}^{T}$ denotes the probability of the distribution of the Gm-APD’s responses or the post probability, in which the superscript means the transpose operation, and the formula $P\left(j,i\right)=P\left(j|i\right)P\left(i\right),i=1,\;\cdots ,\;N,\;j=1,\;\cdots ,\;N$ is satisfied. The target location before the detection ending is completely unknown by the LiDAR system, and the target is possibly located in any bin within the range gate; so, the prior probability is assumed to be a uniform distribution, i.e.,
(5)$$P\left(i\right)=\frac{1}{N},\;i=1,\cdots ,\;N.$$ The probability of the distribution of the Gm-APD’s responses is given in the literature [36]; then, the discretized post probability is
(6)$$\begin{array}{cc} P\left(j\right)= & \left(\frac{N_{s}\Delta t}{\sqrt{2\pi }\sigma }\exp\left(-\frac{{\left(j\Delta t\right)}^{2}}{2\sigma_{}^{2}}\right)+f_{n}\right)\\ & \times \exp\left(-\frac{N_{s}}{2}\left(1+\mathrm{erf}\left(\frac{j\Delta t}{\sqrt{2}\sigma_{}}\right)\right)-f_{n}j\Delta t\right), \end{array}$$
where $\mathrm{erf}\left(\cdot \right)$ is the Gauss error function, and j=1,⋯,N. Substituting Equations (Equation 2), (Equation 5), and (Equation 6) into Equation (Equation 4), the proposed PSE for single-pulse detection is obtained to evaluate the signal quality of the photon-counting LiDAR. Equation (Equation 4) provides a quantitative description of the eliminated uncertainty of the target range after one pulse is received, given the number of signal photons $N_{s}$ and the noise rate $f_{n}$.

### 2.3. PSE for Multiple-Pulse Detection

In photon-counting LiDAR, multiple-pulse accumulation is essential to avoid the detection uncertainty introduced by the inevitable noise. In this case, the mentioned PSE for single-pulse detection cannot describe the influence of the cumulative pulses on the signal quality; so, the PSE needs to be extended for multiple-pulse detection. For a Gm-APD with long deadtime, as the width of the laser pulse is much less than the width of the range gate, the responses in each individual echo pulse are independent from each other. Thus, the forward probability matrix for single-pulse detection is extended into
(7)$$P_{t|\tau }^{K}=\left[\prod_{k=1}^{K}P\left(j_{k}|i_{k}\right)\right],$$
where $i_{k}=1,\;\cdots ,\;N$, $j_{k}=1,\;\cdots ,\;N$, the superscript *K* means *K* accumulation of pulses, and $P_{t|\tau }^{K}$ with the dimension of $N^{k}\times N^{k}$ is the extended forward probability matrix for multiple-pulse detection. The extended forward probability matrix describes the capacity of multiple-pulse statistical detection under the premise of traversing all possibilities of the target location.

Similarly, the extension is operated according to the prior probability and the post probability; that is,
(8)$$P_{\tau }^{K}={\left[\prod_{k=1}^{K}P\left(i_{k}\right)\right]}^{T},$$
and
(9)$$P_{t}^{K}={\left[\prod_{k=1}^{K}P\left(j_{k}\right)\right]}^{T},$$
where $P_{\tau }^{K}$ and $P_{t}^{K}$ denote the extended probability of the target location and that of the distribution of the GM-APD’s responses, respectively. Substituting Equations (Equation 7)–(Equation 9) into Equation (Equation 4), the PSE for multiple-pulse detection is
(10)$$I_{M}=E_{P_{t,\tau }^{K}}\left(\log P_{t,\tau }^{K}-\log P_{\tau }^{K}{\left(P_{t}^{K}\right)}^{T}\right),$$
where $P_{t,\tau }^{K}$ is the extended joint probability. According to information theory, due to the independence of each laser pulse, the PSE for multiple-pulse detection satisfies
(11)$$I_{M}=KI_{S},$$
from which there is a linear relationship between the PSE for multiple-pulse detection and that for single-pulse detection. Equation (Equation 11) describes the quantity of the eliminated uncertainty of the target’s range after multiple-pulse accumulation. Based on the established information transmission model, this work uses the PSE defined in Equations (Equation 4) and (Equation 11) as the statistical metrics to evaluate the signal quality of photon-counting LiDAR; then, a prediction model is proposed to estimate the number of cumulative pulses in high background-noise environments.

## 3. Simulation Analysis

In this section, we describe the simulation and analysis of the characteristics of the PSE. The simulation parameters were as follows: the wavelength of the laser was 532 nm, the width of the laser pulse was 4 ns, the width of a time bin was 64 ps, and the width of the range gate was ∼65.5 ns; the target was located in the 760th time bin.

For single-pulse detection, the characteristics of the PSE evaluation were firstly simulated and compared with the SNR evaluation. The SNR was calculated within the signal duration using the definition in [14]. Figure 1 shows the comparison of the SNR evaluation with the PSE evaluation under various signal and noise conditions, in which the SNR and PSE were both normalized by their maximums. The results in Figure 1a,b show that the SNR and PSE both increased with the number of signal photons. However, in the weak signal region, the SNR decreased slightly with the noise, but the proposed PSE decreased rapidly to a very low level. In photon-counting LiDAR, limited by the Gm-APD’s deadtime, the signal response is affected by all the noise in front of the signal within the range gate, resulting in the nonlinear influence of the noise on the statistical detection, which is depicted by the normalized detection probability. The normalized detection probability is defined as $P_{d,n}=\frac{1-\exp\left(N_{s}\right)}{1-\exp\left(N_{\mathrm{sn}}\right)}$ that indicates the ratio of the signal response probability to the total response probability, with the number of signal photons $N_{s}$ and the number of total echo photons $N_{\mathrm{sn}}$. As shown in Figure 1c,f, in the region of $N_{s}<0.2$, the normalized detection probability decreased obviously with the increase in noise, and according to Figure 1d,e, the proposed PSE was consistent with the normalized detection probability, but the SNR was not. Therefore, the PSE based on the information transmission model was more sensitive to the noise and described the influence of the noise on the statistical detection more objectively.

For multiple-pulse detection, the range accuracy index is discussed as the function of the SNR and the PSE. The definition of the range accuracy is
(12)$$\sigma_{a}=\frac{1}{M}\sum_{i=1}^{M}\left|R_{i}-R_{\mathrm{real}}\right|,$$
where $R_{\mathrm{real}}$ is the true range of the target, $R_{i}$ is the *i*th measured target range, and *M* is the number of repeated ranging experiments. In the simulation, the ranging experiment was realized by a Monte Carlo method, and the target range was estimated by the matched filter method, due to its good anti-noise performance. The range accuracy indicates the mean error between the measured ranges and the true range. It is mainly determined by the uncertainty of the noise distribution (with a variance of $T_{g}^{2}/12$) when there are not enough cumulative pulses, and it is mainly determined by the uncertainty of the signal distribution (with a variance of $\sigma^{2}$) under a large number of accumulated pulses. According to Poisson statistics, the SNR for multiple-pulse detection is easily derived from its definition
(13)$$\gamma_{M}=\sqrt{K}\gamma_{S},$$
where $\gamma_{S}$ and $\gamma_{M}$ denote the SNR for single-pulse detection and multiple-pulse detection, respectively.

Using Equations (Equation 11)–(Equation 13), the results of the range accuracy in the simulation of the SNR evaluation and the PSE evaluation in a weak signal region are shown in Figure 2. Under the same noise conditions, the trends for the SNR and PSE were the same, i.e., the range accuracy decreased with an increasing SNR and PSE, which was a consequence of the accumulation. However, when comparing any two or more noise conditions in the SNR or PSE evaluation, the trends of the difference between the range accuracy curves differed. In the SNR evaluation, the curves of the range accuracy intersected before the range accuracy trend became stable. With the same stable range accuracy, the larger the noise rate was, the better the SNR. More specifically, there always existed a higher noise level, for which the range accuracy curve was located at the right side of the shown curves. It was difficult to choose an SNR as the reference to ensure all the range accuracy under different noise conditions remained near the stable values. In contrast, in the PSE evaluation, the curves of the range accuracy did not intersect before the stable range accuracy. This means that with a stable range accuracy, the lower the noise rate, the larger the PSE, which infers that there is a minimum PSE reference, i.e., the PSE under zero noise, to ensure all range accuracy under different noise conditions remain close to the stable range accuracy. Exploiting this characteristic, we propose a prediction model for estimating the number of cumulative pulses required for high-accuracy ranging for targets in different background environments.

In fact, the results shown in Figure 2 are both the *x*-axis transformations of the curves of the range accuracy varying with the number of cumulative pulses. Figure 2a–c are the nonlinear transformation and unequal-proportion scaling, while Figure 2d–f are the linear transformation and unequal-proportion scaling. According to Equations (Equation 11) and (Equation 13), in multiple-pulse detection, the necessary but insufficient conditions for range accuracy curves intersecting in the SNR evaluation and PSE evaluation are
(14)$$\left\{\begin{array}{cc} \frac{K_{i}}{K_{j}}={\left(\frac{\gamma_{S,j}}{\gamma_{S,i}}\right)}^{2}, & \mathrm{SNR}\;\mathrm{evaluation}\\ \frac{{K^{\prime }}_{i}}{{K^{\prime }}_{j}}=\frac{I_{S,j}}{I_{S,i}}, & \mathrm{PSE}\;\mathrm{evaluation} \end{array},\right.$$
where $K_{i}$ and $K_{j}$ are the number of cumulative pulses when the range accuracy and the SNRs are both equal under any two noise conditions, while ${K^{\prime }}_{i}$ and ${K^{\prime }}_{j}$ are that when the range accuracy and PSEs are both equal. $\gamma_{S,i}$ and $\gamma_{S,j}$ are the SNRs for single-pulse detection under two noise conditions, while $I_{S,i}$ and $I_{S,j}$ are the PSEs for single-pulse detection under two noise conditions, respectively. Since the proposed PSE is not analytic, a numerical analysis was performed on Equation (Equation 14). Based on the results in Figure 1, the conditions in Equation (Equation 14) are visualized in Figure 3, from which $I_{S,j}/I_{S,i}$ was about one to two orders of magnitude larger than ${\left(\gamma_{S,j}/\gamma_{S,i}\right)}^{2}$. Then, we derived that $\left|K_{i}-K_{j}\right|\ll \left|{K^{\prime }}_{i}-{K^{\prime }}_{j}\right|$, which indicated that the difference in the number of cumulative pulses when the PSEs were equal was much larger than that when the SNRs were equal. The results shown in Figure 3 avoided the intersection of the range accuracy curves in the PSE evaluation under different noise conditions before the range accuracy became stable and provided a numerical demonstration of the following prediction model.

## 4. The Prediction Model for Estimating the Number of Cumulative Pulses

According to the simulation analysis, the PSE under zero noise is the reference to ensure the range detection in high background noise remains close to the true range with a low and stable range error. In this section, a prediction model to estimate the number of cumulative pulses is proposed in the following three steps.

(i) Pre-measurement. First, we conducted a pre-measurement to estimate the noise rate and the number of signal photons. Since the signal is not usually located in the front of the range gate, the data in the first *q* bins in the photon-counting histogram were used to estimate the noise rate
(15)$${\hat{f}}_{n}=-\frac{1}{q}\log\left(1-\frac{1}{K_{a}}\sum_{i=1}^{q}y_{i}\right),$$
where $y_{i}$ means the photon-counting data in the *i*th time bin, and $K_{a}$ is the number of cumulative pulses in the premeasurement. The number of signal photons was estimated by regarding the total counts minus the mean noise counts as the signal counts
(16)$${\hat{N}}_{s}=-\log\left(1-\left(\sum_{i=1}^{N}y_{i}-K_{a}\left(1-\exp\left(-N{\hat{f}}_{n}\Delta t\right)\right)\right)/K_{a}\right),$$
where $N=T_{g}/\Delta t$ is determined by the width of the range gate $T_{g}$ and the width of the time bin Δt.

(ii) Estimation of the minimum PSE reference. In a zero noise condition, the response of the Gm-APD must be generated from the signal photons, and the range accuracy must be determined by the signal uncertainty, i.e., the range measured when the Gm-APD generates only one photon count approaches the true range with the stable range accuracy discussed in the simulation. Then, the minimum PSE reference is
(17)$$I_{R}=\frac{1}{1-\exp\left(-{\hat{N}}_{s}\right)}I_{S}\left({\hat{N}}_{s},0\right),$$
where the term $\frac{1}{1-\exp\left(-N_{s}\right)}$ means the number of required cumulative pulses when the Gm-APD generates one photon count, and $I_{S}\left({\hat{N}}_{s},0\right)$ is the PSE for single-pulse detection with the signal photons of ${\hat{N}}_{s}$ and the noise rate of 0.

(iii) Estimation of the number of cumulative pulses. Using the minimum PSE reference and Equation (Equation 11), the number of cumulative pulses required for high-accuracy ranging in high background noise is
(18)$$K_{d}=\frac{I_{R}}{I_{S}\left({\hat{N}}_{s},N{\hat{f}}_{n}\Delta t\right)},$$
where $I_{S}\left({\hat{N}}_{s},N{\hat{f}}_{n}\Delta t\right)$ is the PSE for single-pulse detection with the signal photons of ${\hat{N}}_{s}$ and the noise rate of ${\hat{f}}_{n}$.

The prediction model was experimentally verified. The experimental ranging data were obtained from the photon-counting ranging system established in our previous work [37]. Table 1 lists the key system parameters and pre-measurement setup for the experimental verification. Using the given experimental parameters, the theoretical prediction of the number of cumulative pulses under various signals and noise is shown in Figure 4, which indicates that the cumulative pulses increased quasiexponentially with the noise, and a large number of cumulative pulses was required for high-accuracy ranging in a weak signal and strong noise conditions. Table 2 shows the prediction results in two different experimental noise cases. Referencing the PSE where the Gm-APD generated one photon count in zero noise, the high-accuracy range detection required 22,200 and 7220 cumulative pulses under the background noise of 7.6 MHz and 5.1 MHz, respectively. Figure 5 shows the results of 50 repeated ranging experiments with the estimated numbers, 22,200 and 7220, of cumulative pulses for the two noise cases. The measured ranges in the two cases both concentrated near the true range, with the range accuracy of 4.1 cm and 5.3 cm, respectively, and the standard derivations of 3.8 cm and 3.9 cm, respectively. Figure 5 demonstrates that the proposed PSE prediction model effectively estimated the number of cumulative pulses required for high-accuracy ranging in a MHz-level background.

In addition, the relationship between the range accuracy with the number of cumulative pulses under the two noise cases was explored to test the sampling time cost, as shown in Figure 6. The range accuracy measured after accumulating 22,200 and 7220 pulses under the two noise cases were both close to the stable range accuracy, and comparing the results in Figure 6b,e with Figure 6c,f, the measured ranges had poor performance when there were not enough cumulative pulses. In Case 1, the threshold cumulative number where the range accuracy became stable was about 15,000, as shown in Figure 6a, where the range accuracy was ∼4.4 cm, and the standard deviation was ∼4.5 cm. Similarly, in Case 2, the threshold cumulative number was about 3500, as shown in Figure 6d, where the range accuracy was ∼6.1 cm, and the standard deviation was ∼6.4 cm. It can been seen that the predicted cumulative numbers were increased by 7000 and 3500 extra accumulations, respectively, over the thresholds. The possible reasons are that on the one hand, an estimation error could be introduced by the crude method of estimating the $N_{s}$ and $N_{n}$, and on the other hand, the thresholds are also not precise values as they are the results of multiple averaging, which cannot ensure that the accumulations reach the thresholds in all cases to obtain the same ranging error. In consideration of the performance of the LiDAR sampling time, when the photon-counting LiDAR system uses a laser with low repetition frequency (kHz level), the increasing time cost for one realization of range detection is about several seconds, which greatly reduces the real-time performance of the system. Fortunately, when using a laser at MHz repetition frequency, the increasing time cost is reduced to about several microseconds, which influences the real-time performance only a little. In addition, the extra accumulations provide robustness to eliminate the influence of noise on range measurement of remote targets in background noise. The proposed PSE prediction model is reliable for estimating the number of cumulative pulses required for high-accuracy ranging in high background noise for photon-counting LiDAR with a high repetition frequency. Therefore, the results in Figure 5 and Figure 6 demonstrate that the PSE prediction model is helpful to guide the quasioptimal real-time performance for photon-counting LiDARs working in environments with background noise.

## 5. Conclusions

In conclusion, a PSE-based signal evaluation method was proposed to evaluate the signal quality of the statistical detection for photon-counting LiDAR working in background noise; then, a prediction model for estimating the number of cumulative laser pulses required for high-accuracy ranging was established. First, the detection process of the Gm-APD was developed as an information transmission model, and the PSE was defined to quantitatively describe the eliminated uncertainty of the targets’ range after the echo photons responded to evaluate the signal quality of the photon-counting LiDAR. Then, referencing the PSE of the high-accuracy ranging under the zero noise condition, the number of required cumulative laser pulses for targets’ ranging with background noise was estimated. The simulation analysis showed that the proposed PSE was more sensitive to describing the influence of the noise on the statistical detection, and the minimum PSE reference was obtained under zero-noise conditions to ensure all range detections in the background noise were close to the true range with a low and stable range error. The experiments demonstrated that the prediction model provided a good estimation of the number of required cumulative pulses in various noise conditions. When the signal photons were less than 0.1 per pulse, the range accuracy of 4.1 cm and 5.3 cm were obtained with the estimated number of cumulative pulses in the background noise of 7.6 MHz and 5.1 MHz, respectively. The extra accumulations for one realization of range detection under the estimated number of cumulative pulses only costs several microseconds when using a laser at MHz repetition frequency, but they provide robustness to eliminate the influence of background noise. The PSE evaluation method and the prediction model have the potential to guide the selection of the cumulative number under a satisfactory range accuracy, which is helpful to improve the real-time performance of photon-counting LiDAR and may be applied in LiDAR system design and remote target ranging with background noise.

## Figures and Tables

**Figure 1 entropy-25-00522-f001:**
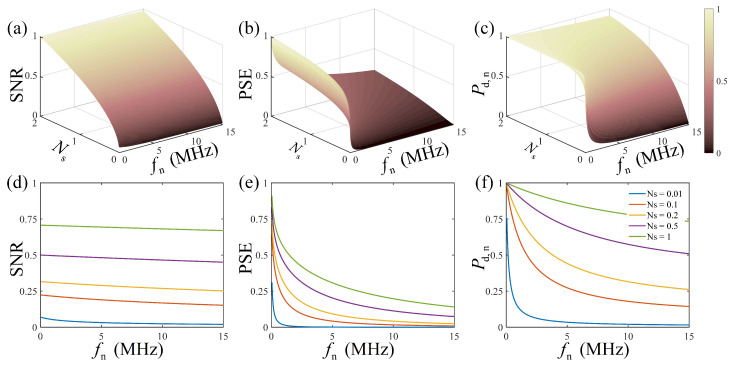
The comparison of the PSE and SNR with a normalized detection probability. (**a**) The SNR under various signal and noise conditions; (**b**) the PSE under various signal and noise conditions; (**c**) the normalized detection probability under various signal and noise conditions; (**d**–**f**) are the results of some selected signal conditions from (**a**–**c**), respectively. The SNR and PSE are both normalized by their maximums.

**Figure 2 entropy-25-00522-f002:**
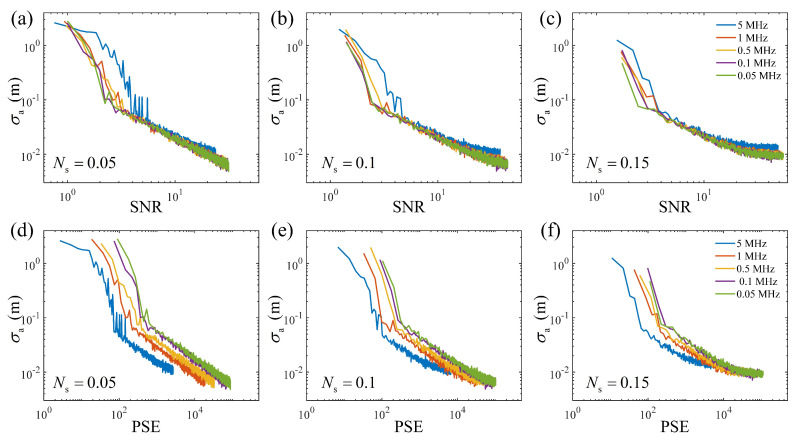
The range accuracy versus the SNR and PSE in multiple-pulse detection. (**a**–**c**) are the results of the SNR evaluation when $N_{s}=0.05,\;0.1,\;\mathrm{and}\;0.15$; (**d**–**f**) are the results of the PSE evaluation when $N_{s}=0.05,\;0.1,\;\mathrm{and}\;0.15$.

**Figure 3 entropy-25-00522-f003:**
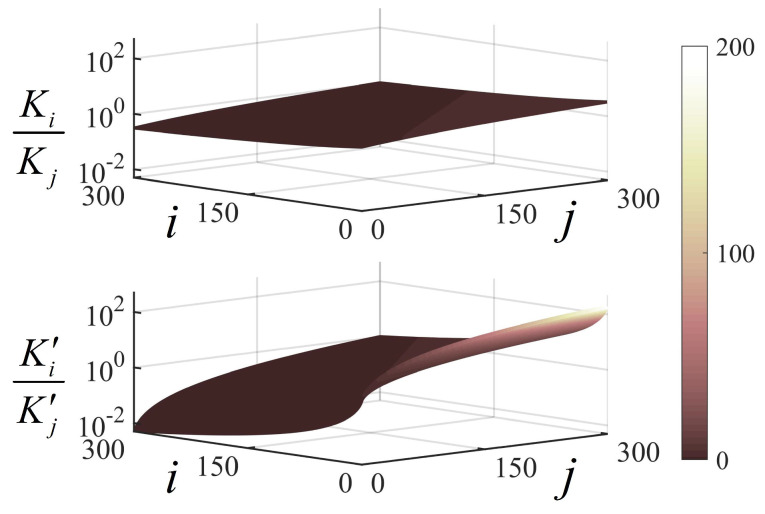
Numerical image of the necessary but insufficient condition for range accuracy curves intersecting when $N_{s}=0.05$.

**Figure 4 entropy-25-00522-f004:**
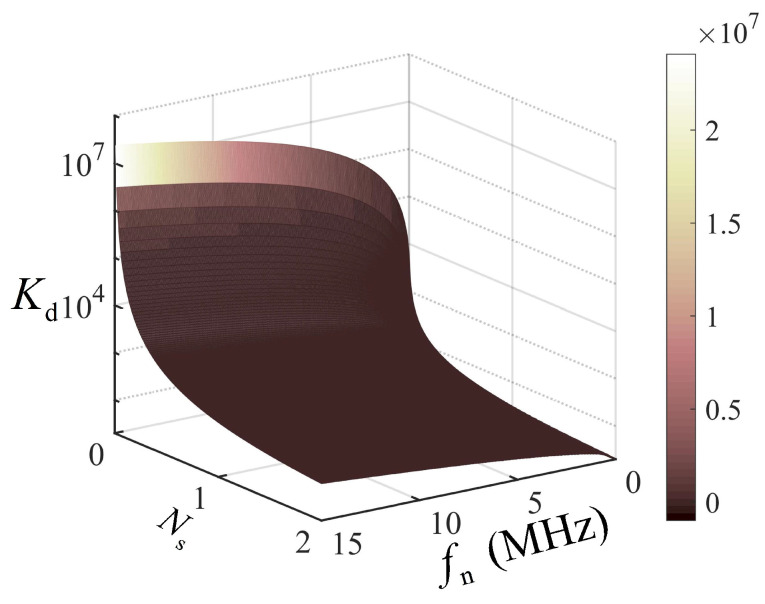
The theoretical prediction of the number of cumulative pulses under the given experimental parameters.

**Figure 5 entropy-25-00522-f005:**
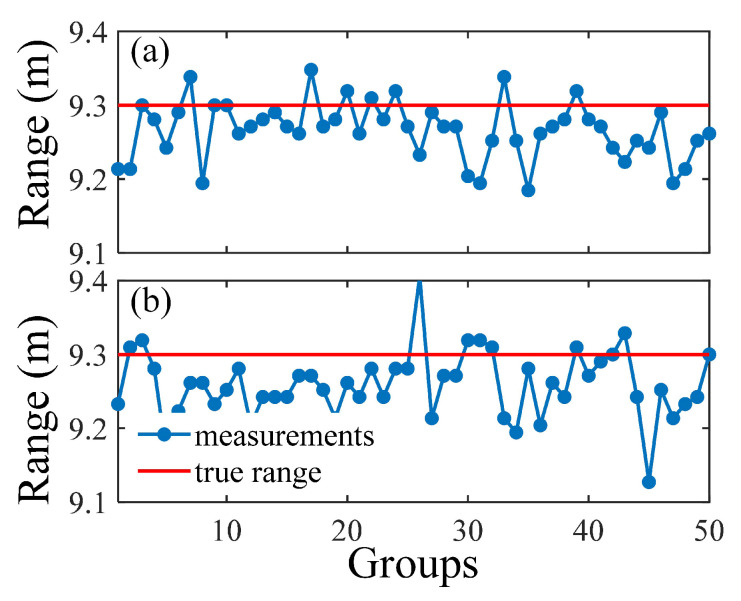
The results of the ranging experiments under the estimated number of cumulative laser pulses. (**a**) The comparison of the measured ranges and the true range with a background noise of 7.6 MHz; (**b**) the comparison of the measured ranges and the true range with a background noise of 5.1 MHz.

**Figure 6 entropy-25-00522-f006:**
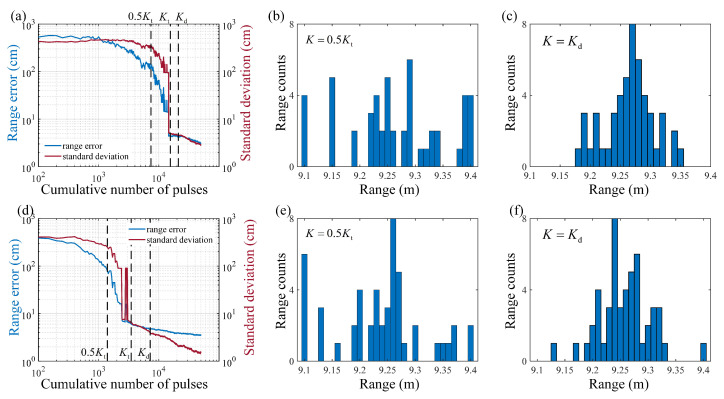
The experimental relationship between the range accuracy and the number of cumulative laser pulses. (**a**,**d**) are the range accuracy and the standard deviation of the measured ranges when the background noise rate is 7.6 MHz and 5.1 MHz, respectively; (**b**,**c**) are the range histograms under the marked number of cumulative pulses in (**a**); (**e**,**f**) are the range histograms under the marked number of cumulative pulses in (**d**). The marked cumulative numbers are the half and the whole of the threshold cumulative number $0.5K_{t}$ and $K_{t}$, respectively, and the predicted cumulative number $K_{d}$. The range histograms depict the distributions of the measured ranges, and the first bin and last bin represent the interval <9.1 m and >9.4 m, respectively.

**Table 1 entropy-25-00522-t001:** Parameters for the experimental verification. $K_{a}$ is the number of cumulative pulses from the premeasurement, and *q* is the number of bins for estimating the noise rate.

	Item	Parameters
System	pulse width	4 ns
	bin width	64 ps
	range gate	102.4 ns
	target range	9.3 m
Pre-measurement	$K_{a}$	1000
	*q*	200

**Table 2 entropy-25-00522-t002:** The estimation results of the prediction model. ${\hat{f}}_{n}$, ${\hat{N}}_{s}$, $I_{R}$, and $K_{d}$ are the estimated noise rate, the estimated number of signal photons, the PSE reference, and the estimated number of cumulative pulses, respectively.

	Case 1	Case 2
${\hat{f}}_{n}$	7.6 MHz	5.1 MHz
${\hat{N}}_{s}$	0.08	0.1
$I_{R}$	1761.0	1422.8
$K_{d}$	22200	7220

## Data Availability

Data available on request due to privacy.

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
