# Peer review of "Prediction of the Number of Cumulative Pulses Based on the Photon Statistical Entropy Evaluation in Photon-Counting LiDAR"

_entropy, 2023, doi:10.3390/e25030522_

Round 1

Reviewer 1 Report

It is an eternal theme that improving the detection precision and accuracy with limited echo photons for lidar. This manuscript proposed a signal quality evaluation method based on photon statistical entropy (PSE) for photon counting lidar.

Two minor suggestions:

(1) Please introduce the concept photon statistical entropy in the introduction and some reference on the photon statistical entropy. And how do you connect the photon counting lidar and the photon statistical entropy theory?

(2) In page 2, formula (1), the lidar echo is assumed as a gaussian function. However, the actual optical flux at time is not perfect gaussian, even the laser pulse is gaussian. Especially for the detection with a resolution of centimeter, the “roughness” will play important role in the echo function in time. What is the influence of the imperfect of the optical flux in the signal quality evaluation methodology based PSE for photon counting lidar?

Reviewer 2 Report

The authors present a new method (photon statistical entropy, PSE) to evaluate the signal quality of photon counting lidar. The new method is stated to outperform the SNR method introduced in 2000 in several aspects like describing the influence of the background noise. The research is well-conducted and article is well written, I only have a few questions:

A) Page 2. line 77: while the reference [24] may be easily reachable, in my opinion it would be interesting to include a short description of N_s.

B) I do not understand why a different range error definition is used (Eq. (12)) and not the standard deviation.

C) Figure 6. Distribution shown on (c) and (f) resembles to Gaussian distribution. Can this be used to quickly evaluate if in a real-world measurement we reached the correct number of pulses?

Round 2

Reviewer 1 Report

  • No more comments. I agreed to publish.

Reviewer 2 Report

The authors answered my questions, I have no further remarks.